🔓 | **Open Peer Review** | Host-Microbial Interactions | Research Article

# Investigating the influence of *Diadematidae* scuticociliatosis on host microbiome composition

Brayan Vilanova-Cuevas,[1] Christopher M. DeRito,[1] Isabella T. Ritchie,[2] Christina A. Kellogg,[3] James S. Evans,[3] Alizee Zimmerman,[4] Stacey M. Williams,[5] Marilyn Brandt,[6] Moriah Sevier,[6] Samuel Gittens Jr.,[6] Kayla A. Budd,[6] Matthew Warham,[7] William C. Sharp,[8] Gabriel A. Delgado,[8] Alwin Hylkema,[9,10] Kimani Kitson-Walters,[11,12] Jean-Pascal Quod,[13] Mya Breitbart,[2] Ian Hewson[1]

**ABSTRACT** Mass mortality of Diadematidae urchins, caused by the *Diadema antillarum* scuticociliatosis Philaster clade (DScPc), affected the Caribbean in spring 2022 and subsequently spread to the eastern Mediterranean, Red Sea, and western Indian Ocean. A key question around Diadematidae scuticociliatosis (DSc), the disease caused by the scuticociliate, is whether the urchin microbiome varies between scuticociliatosis-affected and grossly normal urchins. Tissue samples from both grossly normal and abnormal *Diadema antillarum* were collected in the field during the initial assessment of the DSc causative agent and from an experimental challenge of DScPc culture on aquacultured *D. antillarum*. Specimens were analyzed using 16S rRNA gene amplicon sequencing. Additional abnormal urchin samples were collected from the most recent outbreak site in the western Indian Ocean (Réunion Island). At reference (i.e., unaffected by DSc) sites, *Kistimonas* spp., *Propionigenium* spp., and *Endozoicomonas* spp. were highly represented in amplicon libraries. DSc-affected urchin amplicon libraries had lower taxonomic richness and a greater representation of taxa related to *Fangia hongkongensis* and *Psychrobium* spp. Amplicon libraries of urchins experimentally challenged with the DSc pathogen had some shifts in microbial composition, but *F. hongkongensis* was not a part of the core bacteria in DSc-challenged specimens. DSc-affected *Echinothrix diadema* from Réunion Island showed a similar high representation of *F. hongkongensis* as that seen on Caribbean *D. antillarum*. Our results suggest that DSc alters *Diadematidae* microbiomes and that *F. hongkongensis* may be a candidate bacterial biomarker for DSc in environmental samples. The mechanism driving microbiome variation in host–pathogen interactions remains to be explored.

**IMPORTANCE** The mass mortality of Diadematidae urchins due to *Diadema antillarum* scuticociliatosis (DSc) has had significant ecological impacts, spreading from the Caribbean to the eastern Mediterranean, Red Sea, and western Indian Ocean. This study investigates whether the microbiome of urchins varies between those affected by DSc and those that are not. Using 16S rRNA gene amplicon sequencing, researchers found that DSc-affected urchins had lower taxonomic richness and a greater representation of *Fangia hongkongensis* and *Psychrobium* spp. The findings indicate that *F. hongkongensis* could serve as a bacterial biomarker for DSc in environmental samples, providing a potential tool for early detection and management of the disease. Understanding these microbiome changes is crucial for developing strategies to mitigate the spread and impact of DSc on marine ecosystems.

**KEYWORDS** microbiome, disease, *Diadema antillarum*, scuticociliatosis

D iadematidae scuticociliatosis (DSc) is a condition caused by a scuticociliate within the Philaster clade (abbreviated as DScPc) (1). DSc resulted in mass mortality of

**Peer Reviewers** Colette Feehan, Montclair State University, Montclair, New Jersey, USA; Linda Wegley Kelly, University of California San Diego Scripps Institution of Oceanography, San Diego, California, USA

Address correspondence to Brayan Vilanova-Cuevas, byv3@cornell.edu.

The authors declare no conflict of interest.

See the funding table on p. 11.

*Diadema antillarum* in the Caribbean in spring–summer 2022 (2) and has since been reported in the eastern Mediterranean, Red Sea, and western Indian Ocean (3–5). The rapid geographic spread of this pathogen has inspired questions about how environmental conditions and host microbiome compositional variation affect the DScPc-urchin infection outcome (1–5).

Diadematidae urchins play significant roles in coral reef ecosystems. In the Caribbean, herbivory by *Diadema antillarum* aids in the maintenance of macroalgae, promoting the survival of existing coral and settlement of juvenile coral (6–8). Hence, mass mortality of Diadematidae may cause ecosystem phase shifts in regions where urchins serve similar roles. A previous mass mortality of *D. antillarum*, first reported in 1983, led to the loss of algal-free halos (areas where grazing activity allows for coral settlement [9]), ultimately resulting in a phase shift across the Caribbean from coral to algae that has persisted in the subsequent 40 years (10, 11). Whether the 2022 Caribbean mass mortality and mass mortalities elsewhere will result in similar alteration of the reef and benthic community structures in the short and long term remains to be seen. Moreover, *D. antillarum* recovery post-mass mortality events have been slow taking over a decade until natural recovery was seen on most reefs after the first mortality event in 1983. Currently, it is hypothesized that due to the larger unaffected population of urchins in areas of the Caribbean, the recovery should be substantially faster than before, but this remains to be seen (2).

The sea urchin microbiome, referring to all microorganisms within tissues and on urchin surfaces, may play important roles in host metabolism, digestion, and other processes (12, 13). The composition of Diadematidae microbiomes has been recently assessed by 16S rRNA gene amplicon sequencing (14–16) and these studies have revealed much about the environmental distribution and host taxonomic specificity of constituents. For example, bacterial genera (*Kistimonas*, *Propionigenium*, *Photobacterium*, and *Endozoicomonas*) associated with the spines of *D. antillarum* are also commonly found in reef sediments and on macroalgae (14) and are also found across a broad diversity of sea urchin species including *Echinothrix calamaris*, *Diadema setosum*, *Stomopneustes variolaris*, and *Diadema savignyi* (16, 17). In *Tripneustes gratilla*, which is not within the Diadematidae, *Propionigenium*, and *Photobacterium* are also associated with the epidermis (18). The functional role of genera such as *Propionigenium* sp. and *Photobacterium* sp. within urchin microbiomes has not yet been studied, but in clams, *Kistimonas* spp. aid in metabolism and nutrient acquisition (19). Moreover, outside the Diadematidae, multiple bacterial groups, including the NS10 marine group, *Vibrio* spp., *Pseudoalteromonas* spp., and *Tenacibaculum* spp., have been associated with bald sea urchin disease, which has some gross similarities with DSc (20).

The influence of microorganisms on and within metazoa extends beyond host biology to the broader food web since microbiome constituents may be consumed by bacterivores inhabiting animal surfaces (21). The consumption and release of organic matter by bacterivores promotes the growth of co-occurring heterotrophic bacteria in a relationship known as the microbial loop (22). Bacterivorous ciliates play large roles as part of the microbial loop (23). They can also consume bacterial pathogens of fish and modulate bacterial communities of the reef and its inhabitants (24, 25). Conversely, the host microbiome may modulate infection risk by ciliates since ciliate infection can be correlated to disruptions of the normal microbiome (26), and interruption of invertebrate microbiomes has been hypothesized to result in decreased host health (27). Ciliates can also be pathogens for a variety of metazoa (28–31). Scuticociliates (Oligohymenophorea; Scuticociliatia) cause several marine diseases, primarily resulting in skin lesions (32–35). Some scuticociliates may cause invasive or systemic infections (33, 35), as seen in Diadematidae urchins (1, 33). Interestingly, most pathogenic scuticociliates consume both animal tissues (i.e., histophagy), as well as bacteria and other microorganisms (i.e., bacterivory/grazing) (24). However, scuticociliate impacts (especially from pathogenic taxa) on the structure of prokaryotic communities on and within animal surfaces/tissues have not been widely assessed.

The DScPc scuticociliate (36) invades Diadematidae beginning with spines and spine base tissues, proceeding through their outer body wall, and finally, invading coelomic fluid (1). DScPc pathogenesis is likely a result of histophagy on tissues; however, its exact mechanism of invasion and innate immune (e.g., coelomocyte) evasion is not well understood. Bacteria alone do not cause DSc since <5 µm filtrates of DScPc culture do not induce DSc signs in the experimental challenge (1). However, we hypothesize that DScPc may impact the Diadematidae microbiome structure by consuming microbiome constituents and liberating tissue-derived organic materials. The aim of this study was to describe the bacterial communities associated with grossly normal and DSc-affected specimens of *D. antillarum*. Furthermore, we examined whether differences between abnormal and normal microbiomes were consistent between field-collected and experimentally challenged aquacultured urchins. Finally, we sought to examine whether trends in microbiome composition between abnormal and normal urchins observed in the Caribbean *D. antillarum* specimens were consistent in *Echinothrix diadema* recovered from the western Indian Ocean.

## MATERIALS AND METHODS

### Field sample collection and processing

*D. antillarum* specimens were collected from 24 sites across 10 jurisdictions in the northeastern Caribbean Sea as reported in Hewson et al. (1) (Table S1). Further specimens of abnormal *E. diadema* were obtained from the western Indian Ocean (Réunion Island) in autumn 2023 ($n = 13$). In most jurisdictions, urchins (grossly normal and DSc-affected) were collected at affected sites and at reference sites (well away from the disease).

Specimens were collected by SCUBA divers or snorkelers and transported to the laboratory in ambient seawater prior to dissection. Upon retrieval to the laboratory, urchins were necropsied into several tissue samples using a sterile technique. Coelomic fluid (0.5–1 mL) was retrieved using sterile syringes and needles (37). Spines and spine base tissues were removed using clean pliers. Samples of body wall (4 mm$^2$) were excised using shears, while gonad and digestive tract specimens were removed using sterile forceps after bisecting the test. All tissue samples were preserved in RNALater (Invitrogen, Carlsbad, CA, USA) and transported to the laboratory at Cornell University at room temperature except for samples from Saba and St. John which were frozen in liquid N$_2$ and transported in a dry shipper. All tissue samples ($n = 307$) from Diadematidae urchins were extracted using the Quick-DNA Insect/Tissue kit (Zymo Research, Irvine, CA, USA) per the manufacturer's protocols. Extracted DNA was quantified using PicoGreen fluorescence (Invitrogen).

### Experimental challenge specimen processing

Laboratory-reared specimens of *D. antillarum* challenged with DScPc culture FWC2 as reported previously (1) were also included in this analysis. Twenty-one individuals were housed in aquaria at the University of South Florida College of Marine Science in 5 µm filtered Gulf of Mexico seawater. Ten urchins were exposed to the scuticociliate culture (around ~100 ciliates counted by microscopy), five received filtered (<5 µm) culture to control for the effects of bacteria in the scuticociliate culture (the culture is not axenic), and six urchins received deionized water. The urchins were monitored every 24 h and subject to 50% water changes every 48 h. Signs of disease began after 72–96 h days post inoculation with the ciliate and the experiment was terminated after 7 days. Body wall samples collected from FWC2 challenged ($n = 4$) and control urchins ($n = 3$) were processed as described above for field-collected urchins.

## 16S rRNA gene amplification and read processing

Bacterial and archaeal communities in tissue samples ($n$ = 314) were analyzed using dual-barcoded PCR amplification and sequencing of the V4 region of the 16S rRNA gene (38). Each 40 µL PCR reaction contained 1× PCR master mix (One-Taq Quick-Load 2× Master Mix with Standard; New England Biolabs, Ipswich, MA, USA), 0.125 µM of each barcoded primer (515f; 5′-GTG YCA GCM GCC GCG GTA A3′ and 806r; 5′-GGA CTA CNV GGG TWT CTA AT-3′)(39, 40), and 2 µL of template. Thermocycling conditions followed Apprill et al. (39). The 16S rRNA amplicons were pooled at equal concentrations using the SequalPrep Normalization Plate kit (Invitrogen) and sequenced on an Illumina MiSeq platform (2 × 250 paired end) at Cornell University's Biotechnology Research Center. The 16S rRNA gene amplicon sequences were submitted to EBI (EBI Accession no. PRJEB81238, ERP165086).

Raw sequence data were preprocessed in QIITA (41) under Study ID 14698, where samples were demultiplexed and quality control retained samples with a Phred Score above 30. Sequences were trimmed at 250 nt, and Deblur (42) was used to denoise the sequences, creating an initial amplicon sequencing variants (ASVs) table. Taxonomy was assigned with a 97% identity confidence threshold, using the Silva-138-99-515-806-nb-classifier (V.138) (43), with chloroplast and mitochondrial sequences removed from the ASV table. Data were exported to R for alpha and beta diversity calculations, as well as taxonomic composition analyses using Phyloseq (v.1.46.0) and a graphical summary created using ggplot2 (v.3.5.0) (44, 45). Differences in richness and diversity between DSc-affected and grossly normal urchins were assessed using the Chao1 and Shannon index, respectively (46, 47). Significant differences in alpha diversity were determined using the Kruskal-Wallis test with ggpubr (48). Linear discriminant analysis to determine microbial markers associated with disease states was performed using microbiome Marker v.1.8.0 specifying a linear discriminant analysis score of 6, $P$-value cutoff of 0.05 for Kruskal-Wallis and Wilcoxon test, and counts per million normalization was applied (49). Identification of "core" bacterial groups with the prevalence of at least 0.75% relative abundance and detection in at least 90% of samples of the given sample subset was performed using R packages eulerr (7.0.2), microbiome (v.1.24.0), and microbiome utilities (v.1.00.17) (50–52).

## Phylogenetic analysis of ASVs identified as potential biomarkers of disease

Sequences initially annotated as *Caedibacter taeniosporalis* group and *Psychrobium* spp. were further subject to phylogenetic analysis against close relatives in the NCBI non-redundant database identified by BLASTn (53). The urchin-derived sequences and retrieved close relatives were aligned in MEGA11 with MUSCLE using the UPGMA cluster method and a minimum diagonal length of 24 (54). This resulted in a 251 bp alignment for both phylogenetic analyses. The best distance models were determined using the function Find DNA/Protein Best Model and phylogenetic trees were built using a maximum likelihood, with Kimura-2 parameter and uniform rates among sites for sequences assigned to *Psychrobium* spp. and Kimura-2 and gamma-distributed rates among sites for sequences assigned to the *C. taeniosporalis* group. Trees were constructed following the maximum likelihood-heuristic method and neighbor-joining interchange and bootstrapped 100 times.

## RESULTS

### Characterization of tissue microbiome variation within reference *D. antillarum*

Our analysis permits the identification of normal constituents of the *D. antillarum* microbiome and how it varies between specimens, across tissue types, and between sites. 16S rRNA gene amplicon libraries prepared from the external tissues of grossly normal (i.e., reference) *D. antillarum* (spines and body wall, including test) had a large representation (i.e. dominant ASVs considered those above 5% relative abundance)

of ASVs assigned to *Kistimonas* spp., *Propionigenium* spp., and *Endozoicomonas* spp. (Fig. 1). Internal tissues and coelomic fluid amplicon libraries had different dominant ASVs, including those assigned to *Acinetobacter* spp., *Brevundimonas* spp., *Lactococcus* spp., *Pedobacter* spp. in coelomic fluid, *Photobacterium* spp. in gonads, and *Candidatus hepatoplasma* spp., *Spirochaeta* 2 spp., and *Sulfurimonas* spp. in digestive tracts. *Roseimarinus* spp. was shared between all tissue types. ASVs assigned to *Kistimonas* spp. were shared between external tissues and coelomic fluid, while *Propionigenium* spp. was also recovered from gonads and the digestive tract (Fig. 1).

## Comparison of grossly normal and DSc-affected urchin microbiomes in the Caribbean

The sampling strategy employed for field-collected urchins included reference (i.e., away from DSc-affected sites), grossly normal at DSc-affected sites (normal-AS), and DSc-affected urchins. Despite appearing healthy, NAS is grouped separately from reference urchins because these specimens may have had preclinical (i.e. before developing gross DSc signs) infection which may also have harbored altered microbiomes (1). The alpha diversity (Chao1) of body wall (reference-DSc affected, $P = 0.01$, $n = 52$), gonads (reference-normal-AS, $P = 0.02$, $n = 40$; reference-DSc-affected $P = 0.02$, $n = 39$), and intestines (reference-DSc-affected, $P = 0.01$, $n = 39$) amplicon libraries were less rich (Chao1) in DSc-affected urchins compared to normal-AS or reference libraries (Fig. 2A). However, there was no significant difference in diversity, based on Shannon diversity index and a Kruskal-Wallis statistical test, between any amplicon libraries prepared from different tissues (Fig. 2B). Beta diversity (i.e., comparing microbial community

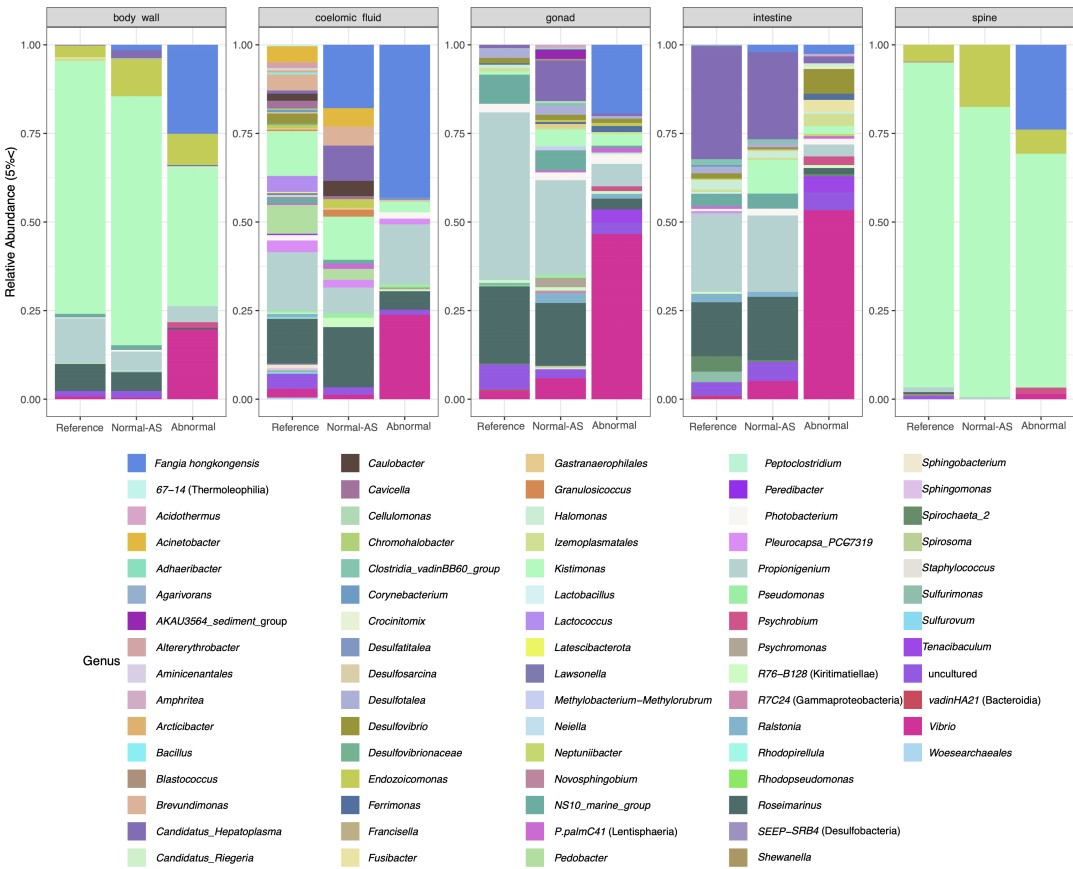

**FIG 1** Taxonomic assessment of bacterial genera from field-collected samples that are above 5% relative abundance. The x-axis represents whether samples were grossly normal at reference site (reference), grossly normal at affected site (normal-AS), or DSc-affected (abnormal). This represents the average abundance across samples pertaining to the same sample type and disease state, and individual sample abundances are also detailed in Fig. S1.

structures) of amplicon libraries was likewise not significantly different based on disease state (normal-AS/DSc-affected/reference) nor source jurisdiction. However, there was significant dispersion based on the tissue type (Adonis $P = 0.01$) (Fig. 2C).

Normal-AS and DSc-affected urchin amplicon libraries had a greater representation of ASVs assigned to the *Caedibacter taeniosporalis* group (Fig. 1). Phylogenetic assessment indicated that these ASVs were most closely related to *Fangia hongkongensis*, and a few were most closely related to *Cysteiniphylum* spp. (Fig. S2). ASVs assigned to *Psychrobium* spp. were most closely related to *Psychrobium conchae* (Fig. S3). These ASVs were broadly conserved across all tissue types of DSc-affected and NAS urchins but had little to no representation in reference urchin amplicon libraries across all tissue types. Linear discriminant analysis of effect size suggests that the *F. hongkongensis* ASVs are a possible bacterial biomarker of DSc in intestines and coelomic fluid. Additionally, *Psychrobium* spp. is a putative biomarker for DSc in the body wall (Fig. 3A). In contrast, ASVs assigned to *Propiniogenium* spp. comprised a greater proportion of reference and normal-AS urchin libraries than of DSc-affected urchin libraries (Fig. 1). Reference urchin amplicon libraries had a significantly higher representation of ASVs assigned to *Propionegenium* spp. and NS10 marine group bacteria (*Cryomorphaceae*) in gonads, and ASVs assigned to *Roseimarinus* spp. in spines, when compared to amplicon libraries from normal-AS and DSc-affected urchins (Fig. 3A) (Kruskal-Wallis and Wilcoxon test $P$-value < 0.05, $n = 292$). Core taxonomic units, which occurred across tissue types and comprised a large proportion of amplicon libraries, were ASVs assigned to the family *Vibrionaceae* and the genera *Propionigenium* spp. and *Kistimonas* spp. in reference and normal-AS urchins, and ASVs assigned to *F. hongkongensis* in DSc-affected tissue amplicon libraries (Fig. 3B).

## Microbiome variation in experimentally challenged *D. antillarum*

Comparing microbiome changes within an experimental setting allows us to better understand if changes in the microbial community directly correspond to that which happens in the environment and whether it influences disease outcome. From the body wall samples collected at the end of the challenge experiment from experimentally challenged, aquaculture-raised *D. antillarum*, seven generated sufficient 16S rRNA gene library coverage for downstream analysis to investigate differences in the microbial communities of those infected with FWC2 compared to controls. This resulted in three controls and four libraries from urchins challenged with the FWC2 culture. The amplicon libraries prepared from controls and FWC2-challenged *D. antillarum* were not significantly different in terms of diversity (Shannon diversity index), based on a Kruskal-Wallis test ($P = 0.11$, $n = 7$) (Fig. 4A). Similarly, community structures were no different between control and challenged specimens (Fig. 4B). Taxonomic assignment of sequences that surpassed 1% relative abundance within each FWC2-challenged specimen library included ASVs assigned to *Photobacterium* spp., *Carboxylicivirga* spp., *Halodesulfovibrio* spp., *Vibrio* spp., and *F. hongkongensis* (Fig. 4C). Furthermore, amplicon libraries prepared from normal urchins had greater representation of ASVs assigned to *Caldalkalibacillus* spp., *Crocinitomix* spp., *Desulfobacter* spp., *Desulfobulbus* spp., *Desulfoarcina* spp., *Draconibacterium* spp., *Halodesulfovibrio* spp., *Woesearchaeales* spp., and *Kistimonas* spp. (Fig. 4C). While the overall community structure (Fig. 4A and B) does not show significant differences, there are differences in the abundance of the previously mentioned taxa. The controls from the challenge experiment, as well as field-collected reference and normal-AS urchins, consistently contained ASVs assigned to *Kistimonas* spp., *Propionigenium* spp., and within the *Vibrionaceae* family, suggesting these are hallmark taxa of grossly normal specimens (Fig. S4A). However, field reference samples had a high representation of ASVs which was most similar to *Ferrimonas* spp., while grossly normal experimental urchin amplicon libraries had a wide representation of *Ralstonia* spp. (Fig. S4A). A comparison of DSc-affected body wall samples from the challenge experiment and DSc-affected field-collected body wall samples indicated some consistencies in compositional microbiome shifts. Field-collected DSc-affected samples show high prevalence and detection of *F. hongkongensis* and *Kistimonas* spp., while experimentally challenged samples show

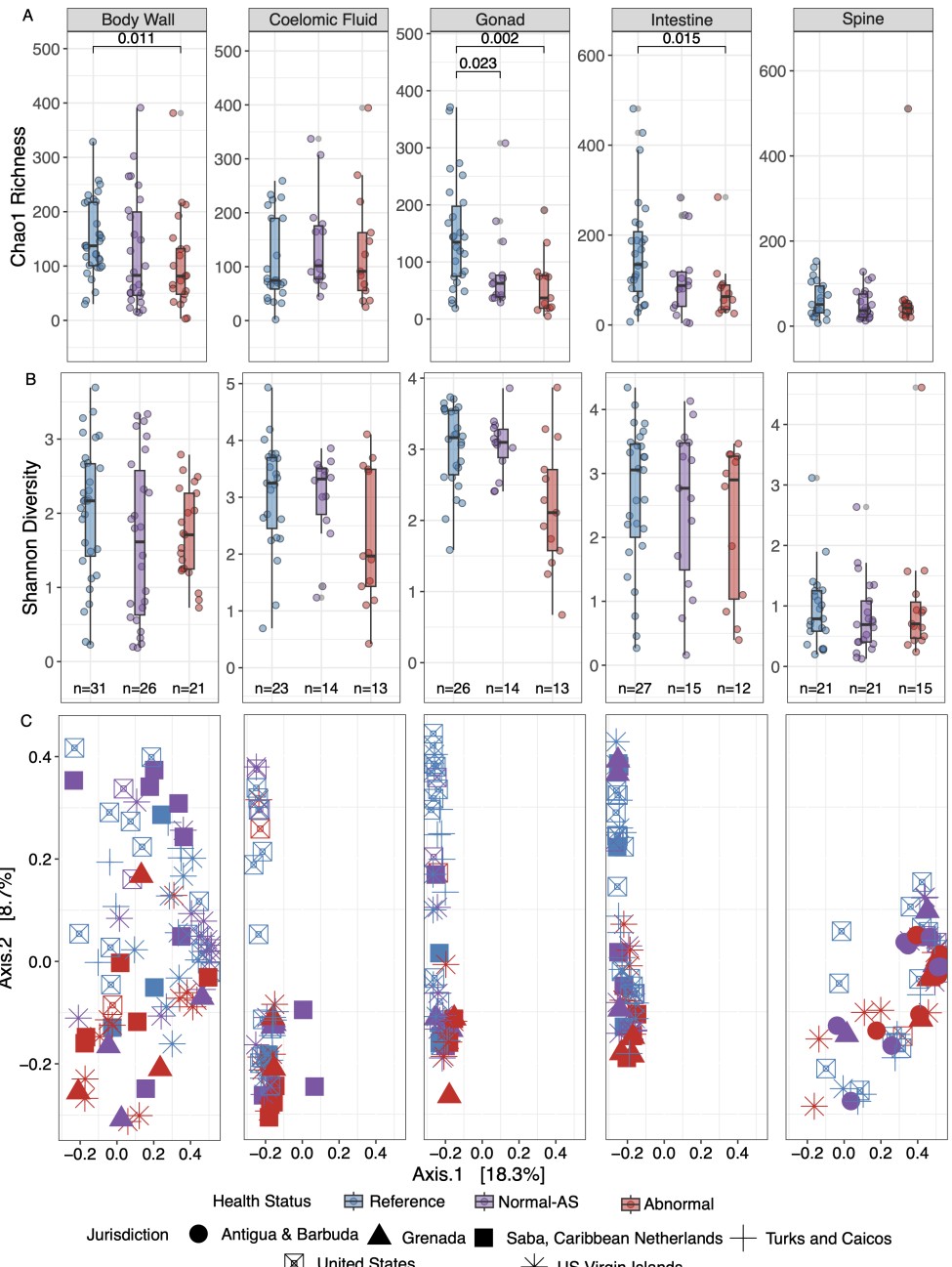

**FIG 2** Alpha and beta diversity of microbial communities by tissue type and jurisdiction from field-collected samples: (A) microbial richness (Chao1 richness index); (B) microbial diversity (Shannon diversity metric) significant differences for both A and B were assessed with a Kruskal Wallis test; *n* = 292); and (C) beta diversity (Bray–Curtis dissimilarity index) visualized by PCoA.

higher prevalence and detection of *Carboxylicivirga* spp. (Fig. S4B). While *F. hongkongensis* was not part of the core microbial community of the experimental DSc-challenged urchins due to not being as prevalent, taxonomic composition assessment shows they at least surpass 1% relative abundance and are present within the challenged urchins.

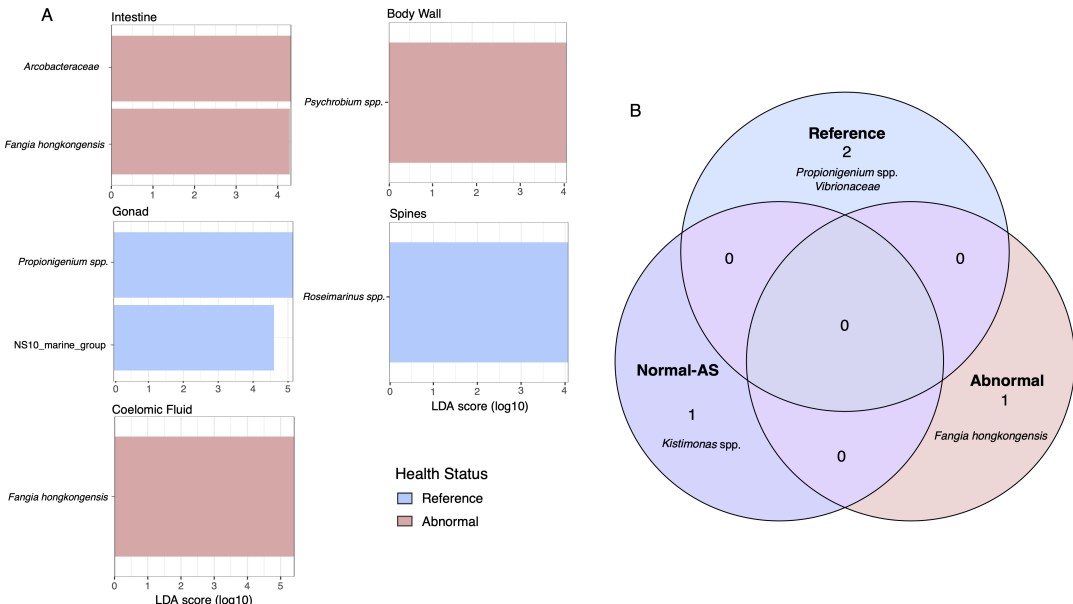

**FIG 3** (A) Linear discriminant analysis effect size showing significantly different (*P* < 0.05, *n* = 292) taxa by tissue state. (B) Core bacterial amplicon sequence variants (ASV) associated with health status in field-collected samples of *Diadema antillarum* across all tissue types. "Reference" = grossly normal urchins collected from sites unaffected by DSc outbreaks; "Normal at Affected Site" = grossly normal urchins collected from sites affected by DSc outbreak; "Abnormal" = urchins that have confirmed DSc infection. Core bacteria are defined by 0.75% relative abundance and 90% prevalence across the samples.

## Comparison of microbiomes of DSc-affected urchins from the Caribbean and grossly abnormal urchins from the western Indian Ocean

Grossly abnormal *Echinothrix diadema* from the western Indian Ocean, collected offshore of Réunion Island, had similar abnormal biomarkers as DSc-affected urchins from the Caribbean. Amplicon libraries, prepared from coelomic fluid and spines (*n* = 13), had a high representation of *F. hongkongensis* and *Psychobium* spp. (Fig. 5A), both genera were identified as markers of DSc in body wall and spine samples from Caribbean *D. antillarum* specimens (Fig. 3A). While the amplicon libraries prepared from DSc-affected urchins from Réunion Island show a similarly high representation of sequences from disease-associated genera, we did not survey grossly normal specimens from this location. Core bacterial community comparison between coelomic fluid samples from abnormal urchins from both locations revealed that out of the two identified biomarker taxa of DSc, *F. hongkongensis* is the only one that surpassed the threshold of detection and prevalence in both locations (Fig. 5B).

## DISCUSSION

Our results suggest that specific bacterial taxa are associated with DScPc infection in the Caribbean and grossly abnormal urchins from Réunion Island with DSc and that urchins experiencing DSc have broadly different microbiome structures than reference specimens. We speculate that some part of this variation may be imparted by the introduction of bacteria associated with the DScPc ciliate itself, but the differences could also result from the consumption of common microbiome constituents, through antagonistic interactions with DScPc-associated bacteria, or through enrichment of organic matter from decaying tissues that selects for copiotrophic taxa. Moreover, DSc could also impact microbial community structures due to disease interactions with urchin immunity, considering invertebrates' innate immunity plays a role shaping its symbiotic constituencies (55). A consistent finding among field-collected and experimentally challenged abnormal urchins was the prominence of two genera within the *Francisellaceae*, *Fangia hongkongensis,* and *Cysteinophilium* sp. *F. hongkongensis* was first isolated from seawater samples collected in Hong Kong and possesses ubiquinone-8 (56), as do all cultivated

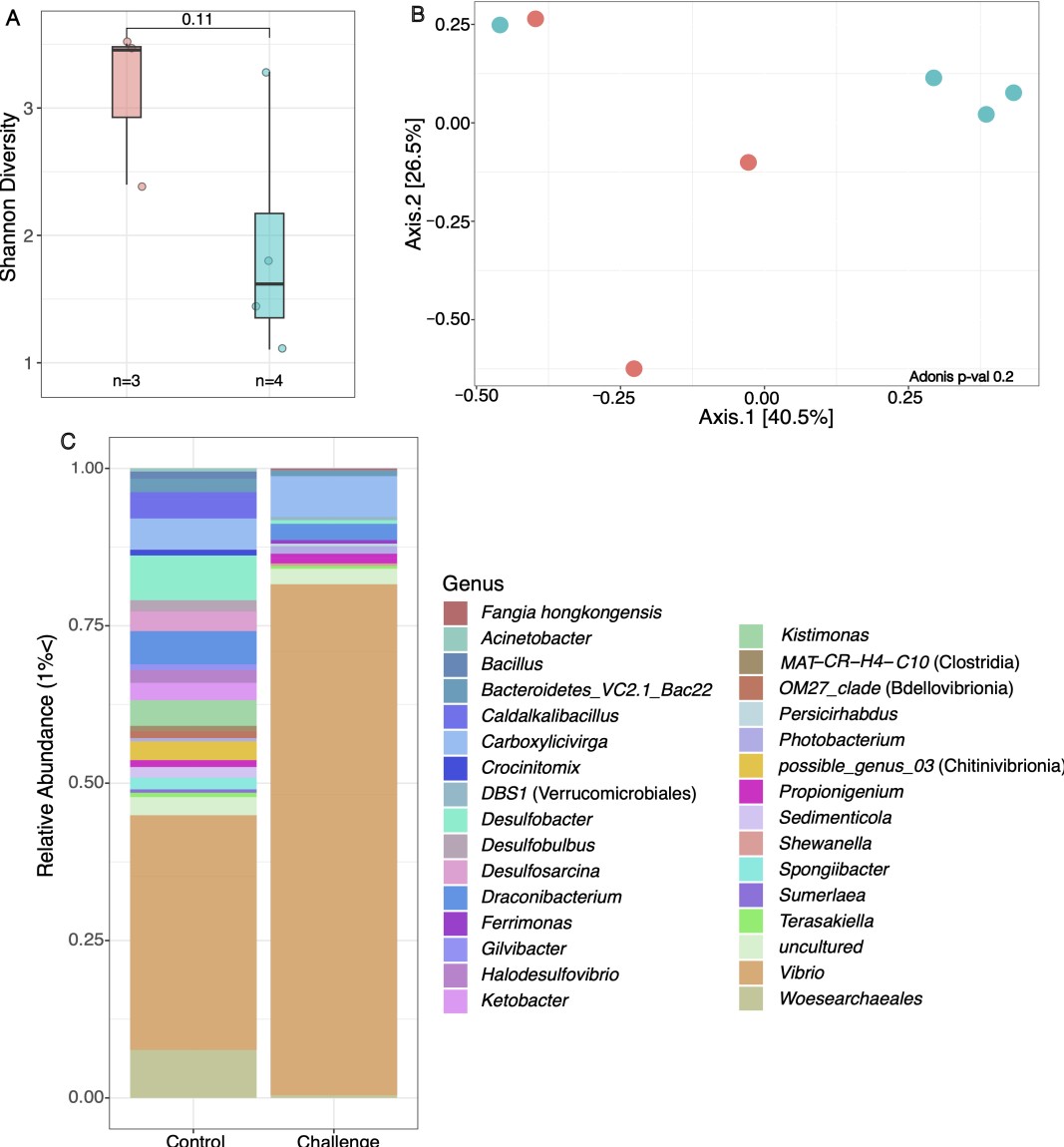

**FIG 4** Microbial community differences in body wall samples of the *D. antillarum* challenge experiment showing diversity assessed by (A) alpha diversity analysis (Shannon diversity index) and significant differences were assessed with a Kruskal-Wallis test; $P > 0.05$, $n = 7$); (B) community structure (Bray–Curtis dissimilarity index) and PCoA visualization; and (C) broad taxonomic composition considering only taxa that surpass 1% relative abundance. Red bars and symbols in panels A and B indicate control specimens, while blue bars and symbols indicate urchins challenged with the DScPc culture FWC2.

members of the *Franscicellaceae* (57). Ubiquinone-8 is an important electron transfer mediator that allows for aerobic respiration in anoxic conditions and aids in mediating oxidative stress in some bacterial species (58). If ubiquinone-8 is used by the *F. hongkongensis* ASV, it could indicate that DSc is associated with lower oxygen conditions within and around affected urchins (59, 60). Moreover, *F. hongkongensis* was also present in DSc-affected *E. diadema* coelomic fluid and spine samples collected from Réunion Island. Other members of the *Francisellaceae* family are symbionts within *Euplotes* sp. and *Paramecium* sp. ciliates. In *Paramecium* spp., a symbiotic relationship with the related *Caedibacter teanospiralis* results in a fitness advantage within *Paramecium* populations, allowing symbiont-bearing ciliates to kill symbiont-lacking cells (61, 62). While the relationship between the ASVs closely related to *F. honkongensis* and DScPc is unknown, our observations suggest that it may be a tight association with DScPc during infection in field settings.

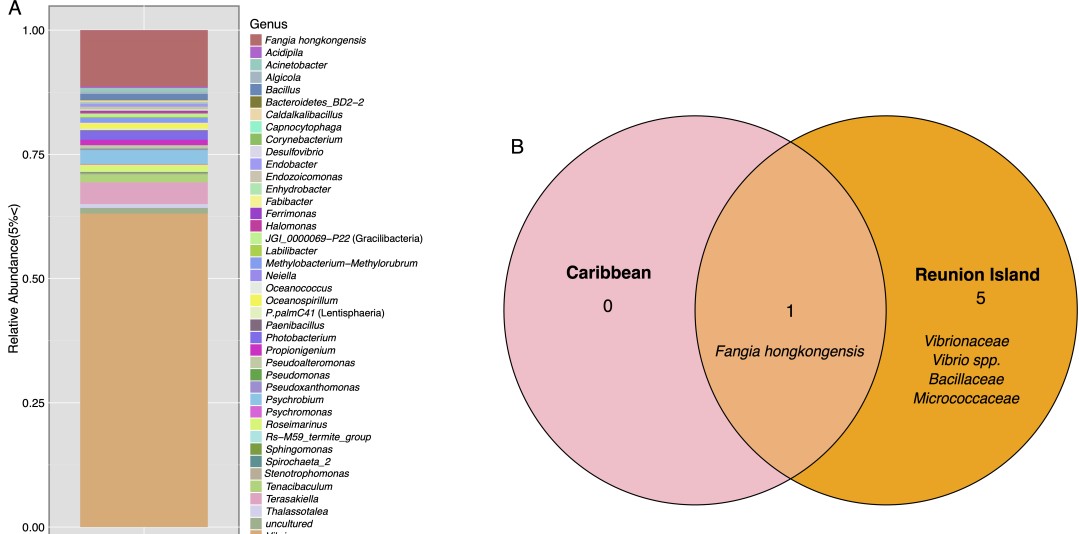

**FIG 5** (A) Taxonomic composition of bacterial taxa comprising >1% relative abundance in abnormal specimens of *Echinothrix diadema* coelomic fluid and spines (collected in the western Indian Ocean, Réunion Island, France). (B) Core bacterial ASVs surpassing 90% prevalence and 0.75% relative abundance in abnormal *Diadema antillarum* (Caribbean) and *E. diadema* (Réunion) coelomic fluid.

Another identified marker of disease was *Psychrobium* sp., which can be highly prevalent in abnormal marine fish, bivalves, and sponges (63–65). However, this biomarker was not as highly represented in the challenge experiment samples when compared to field-collected urchins. Considering the loss of spines and tissue degradation that occurs with DSc, we speculate that this ASV thrives on liberated organic material from decaying tissues and hence represents an opportunist that recruits from surrounding seawater.

Here, we define core bacterial groups as ASVs that surpass 0.75% relative abundance and 90% prevalence across the samples. These ASVs would make up a consistent part of the microbial community across variable habitats and geographic scales. In tropical *Echinometra* spp., bacteria are hypothesized to recruit to the host from surrounding seawater, as there is little evidence of vertical transmission due to different microbiome compositions between gonad and larval microbiomes (66). Comparison of core bacterial communities of hatchery-raised *D. antillarum* in experimental settings to those in field-collected (reference) specimens herein revealed the conservation of core bacteria and the acquisition of *Ralstonia* spp. not seen in the core taxa of field-collected reference samples. This supports that the core microbial community of *D. antillarum* undergoes only minor alterations during containment in filtered seawater aquariums. However, there were differences in non-core ASVs between experimental and field-collected specimens. For example, grossly normal control urchins from the challenge experiment had a large representation of *Desulfobacter* spp., *Desulfobulbus* spp., *Desulfoarcina* spp., and *Halodesulfobacter* spp., all of which are anaerobic and may reduce sulfate to sulfide (67, 68), which can accumulate in saltwater tanks (69). Notably, the urchins that developed DSc during the experiment bore the biomarker *F. hongkongensis*, though it was not abundant enough to be classified as part of the core taxa. This suggests that their abundance might not be tied to DSc pathology.

## Conclusion

Our comprehensive survey revealed distinct microbial communities associated with different tissues of *D. antillarum* and identified bacteria that occurred in both normal and abnormal urchin microbiomes. Reference urchins bore ASVs most similar to *Kistimonas*, *Propionigenium*, and *Roseimarinus* across tissues and jurisdictions, suggesting that they are selected by biotic and abiotic factors of the host. In contrast, DSc-affected

urchins showed a significant reduction in bacterial richness, with *F. hongkongensis* and *Psychrobium* spp. serving as potential biomarkers of DSc in field specimens. Furthermore, these biomarkers of DSc were conserved beyond Caribbean *D. antillarum*, also being found in the urchin species *Echinothrix diadema* in the western Indian Ocean. Our findings underscore that the microbiome disruption by DSc is consistent across broad environmental expanses. Future research on the mechanistic role of identified disease-associated bacteria and DScPc would help resolve its contribution to disease, if any.

## ACKNOWLEDGMENTS

We acknowledge Sarah Von Hoene, Tom Wijers, and David Marancik for their sample collection efforts.

This work was supported by a grant from the Atkinson Center for Sustainable Futures Rapid Response Fund and National Science Foundation grant OCE-2049225 to I.H.; the Von Rosenstiel Fellowship and Von Rosenstiel Innovation Fund for Marine Science to I.T.R.; and NSF Graduate Research Fellowship Program awards to I.T.R. (#2136515) and B.V.-C. (#1650441). J.S.E. and C.A.K. were supported by the U.S. Geological Survey Ecosystems Mission Area Biological Threats and Invasive Species Research Program. Any use of trade, firm, or product names is for descriptive purposes only and does not imply endorsement by the U.S. Government.

## AUTHOR AFFILIATIONS

[1]Department of Microbiology, Cornell University, Ithaca, New York, USA

[2]College of Marine Science, University of South, St. Petersburg, Florida, USA

[3]U.S. Geological Survey St Petersburg Coastal and Marine Science Center, St. Petersburg, Florida, USA

[4]Turks and Caicos Reef Fund, Providenciales, Turks and Caicos Islands

[5]Institute for Socio-Ecological Research, Cabo Rojo, Puerto Rico, USA

[6]Center for Marine and Environmental Studies, University of the Virgin Islands, Saint Thomas, Virgin Islands, USA

[7]Department of Planning and Natural Resources, Virgin Islands Government, Christiansted, Virgin Islands, USA

[8]Florida Fish and Wildlife Conservation Commission, Fish and Wildlife Research Institute, Marathon, Florida, USA

[9]Van Hall Larenstein University of Applied Sciences, Leeuwarden, Netherlands

[10]Marine Animal Ecology Group, Wageningen University, Wageningen, Netherlands

[11]Caribbean Netherlands Science Institute, St. Eustatius, Caribbean, Netherlands

[12]NIOZ Royal Netherlands Institute for Sea Research, Oranjestad, Caribbean, Netherlands

[13]ARVAM, c/o association Technopole de la Réunion, Sainte Clotide, Réunion, France

## AUTHOR ORCIDs

Brayan Vilanova-Cuevas http://orcid.org/0000-0001-6075-3336
Christina A. Kellogg http://orcid.org/0000-0002-6492-9455
Mya Breitbart http://orcid.org/0000-0003-3210-2899
Ian Hewson http://orcid.org/0000-0001-9612-2026

## FUNDING

| Funder | Grant(s) | Author(s) |
| --- | --- | --- |
| National Science Foundation (NSF) | OCE-2049225 | Ian Hewson |
| NSF \| National Science Foundation Graduate Research Fellowship Program (GRFP) | 1650441 | Brayan Vilanova-Cuevas |
| NSF \| National Science Foundation Graduate Research Fellowship Program (GRFP) | 2136515 | Isabella T. Ritchie |

| Funder | Grant(s) | Author(s) |
|---|---|---|
| Von Rosenstiel Fellowship and Von Rosentiel Innovation Fund for Marine Science | | Isabella T. Ritchie |
| CU \| David R. Atkinson Center for a Sustainable Future, Cornell University (ACSF) | | Ian Hewson |
| U.S. Geological Survey Ecosystems Mission Area Biological Threats and Invasive Species Research Program | | Christina A. Kellogg |
| U.S. Geological Survey Ecosystems Mission Area Biological Threats and Invasive Species Research Program | | James S. Evans |

## AUTHOR CONTRIBUTIONS

Brayan Vilanova-Cuevas, Conceptualization, Data curation, Formal analysis, Investigation, Methodology, Visualization, Writing – original draft, Writing – review and editing | Christopher M. DeRito, Investigation, Writing – review and editing | Isabella T. Ritchie, Investigation, Writing – review and editing | Christina A. Kellogg, Investigation, Resources, Supervision, Writing – review and editing | James S. Evans, Investigation, Resources, Writing – review and editing | Alizee Zimmerman, Resources, Writing – review and editing | Stacey M. Williams, Resources, Writing – review and editing | Marilyn Brandt, Resources, Writing – review and editing | Moriah Sevier, Resources, Writing – review and editing | Samuel Gittens Jr., Resources, Writing – review and editing | Kayla A. Budd, Resources, Writing – review and editing | Matthew Warham, Resources, Writing – review and editing | William C. Sharp, Resources, Writing – review and editing | Gabriel A. Delgado, Resources, Writing – review and editing | Alwin Hylkema, Funding acquisition, Resources, Writing – review and editing | Kimani Kitson-Walters, Resources, Writing – review and editing | Jean-Pascal Quod, Investigation, Resources, Writing – review and editing | Mya Breitbart, Funding acquisition, Investigation, Project administration, Resources, Writing – review and editing | Ian Hewson, Conceptualization, Funding acquisition, Investigation, Methodology, Project administration, Resources, Supervision, Writing – original draft, Writing – review and editing

## DATA AVAILABILITY

Raw sequence data were deposited to EBI under accession numbers PRJEB81238 and ERP165086. Processed sequence data are available in QIITA under project ID 14698.

## ADDITIONAL FILES

The following material is available online.

### Supplemental Material

**Supplemental material (mSystems01418-24-s0001.docx).** Supplemental figures and table.

### Open Peer Review

**PEER REVIEW HISTORY (review-history.pdf).** An accounting of the reviewer comments and feedback.

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
