## [Reviewer comments · mSystems]

Investigating the influence of Diadematidae scuticociliatosis on host microbiome composition

Brayan Vilanova-Cuevas, Christopher DeRito, Ian Hewson, Isabella Ritchie, Christina Kellogg, Alizee Zimmerman, Stacey Williams, Marilyn Brandt, Moriah Sevier, Samuel Gittens Jr, Kayla Budd, Matthew Warham, William Sharp, Gabriel Delgado, Alwin Hylkema, Mya Breitbart, James Evans, Kimani Kitson-Walters, and Jean-Pascal Quod

Corresponding Author(s): Brayan Vilanova-Cuevas, Cornell University

Review Timeline:

Submission Date:	October 22, 2024
Editorial Decision:	December 28, 2024
Revision Received:	January 4, 2025
Accepted:	January 21, 2025

Editor: Jean-Baptiste Raina

Reviewer(s): Disclosure of reviewer identity is with reference to reviewer comments included in decision letter(s). The following individuals involved in review of your submission have agreed to reveal their identity: Colette Feehan (Reviewer #1); Linda Wegley Kelly (Reviewer #2)

Transaction Report:

DOI: <https://doi.org/10.1128/msystems.01418-24>

Re: mSystems01418-24 (Investigating the influence of Diadematidae scuticociliatosis on host microbiome composition)

Dear Dr. Brayan Vilanova-Cuevas:

Revision Guidelines

Sincerely,
Jean-Baptiste Raina
Editor
mSystems

Reviewer #1 (Comments for the Author):

Overview

The study investigates the impact of *Diadema antillarum* scuticociliatosis (DSc), caused by the Philaster clade scuticociliate, on the microbiomes of Diadematidae sea urchins following mass mortality events in the Caribbean (Spring 2022) and subsequent

outbreaks in the Eastern Mediterranean, Red Sea, and Western Indian Ocean. Using 16S rRNA gene amplicon sequencing, microbiome composition was compared between grossly normal and DSc-affected urchins from field samples and experimental challenges. Results show that unaffected urchins from reference sites have different representation and taxonomic richness of species than DSc-affected urchins. The findings suggest that DSc alters Diadematidae microbiomes. Some clarification is requested below.

Specific Comments

Abstract: It is perhaps too early to state that *Fangia hongkongensis* could be useful for early detection of *Diadema antillarum* scuticociliatosis as it is unknown at what stage of infection these bacteria colonize the urchin host.

Lines 65-73: Additional interesting context for this disease is the recent efforts to understand factors limiting population recovery of this important grazer; one potentially relevant factor being the ongoing presence of a disease agent in populations.

Lines 101-103: Also, the reverse: might bacterial communities affect risk of infection with ciliates?

Lines 126-137: Why was sterile technique not applied across all types of samples collected?

Lines 290-299: Which hypothesis or combination of hypotheses do you find most likely?

Reviewer #2 (Comments for the Author):

Overall comments:

The study by Vilanova-Cuevas et al. describes the microbiomes associated with normal and abnormal (scuticociliatosis affected) urchins. The dataset represents a comprehensive sample collection of grossly normal and scuticociliatosis infected urchins (~300) from two major geographies, the Caribbean and the Indian Ocean. The study identified some interesting taxa that may serve as indicator species and point towards a functional understanding of the pathology. In my opinion, the study will make a good contribution to the field and is appropriate for the scope of mSystems. While the microbiome analysis was rigorous, there are several details that need to be included in the text and some of the graphics should be reconfigured. Detailed below are comments to improve both quality and clarity of the manuscript.

Specific comments:

Abstract:

Line 31, DScPc is never defined. Probably easiest to add the abbreviation to line 26. Also, the abstract uses abbrev. DaScPc while other sections use DScPc.

Introduction:

Lines 59-60, For consistency, I recommend revising to: "Diadematidae scuticociliatosis (DSc) is a condition caused by a scuticociliate within the Philaster clade (abbreviated as DScPc)."

Methods:

Lines 179-190. Use this section to describe the name changes from *C. taeniosporalis* to *F. hongkongensis*, etc. Then refrain from mentioning *Caedibacter* in the results text or figure captions because this is confusing.

Results:

Lines 196, 207, 231, etc. Capitalize Reference when referring to that specific sample type and fix throughout.

Lines 197-202. When you refer to ASVs, I believe this sequence grouping would represent one individual in the population, a sp. not spp. (which denotes several different species/strains within the genus). Unless you clustered sequences at 97% identity (OTUs), then it would be more appropriate to refer to the taxonomic group as spp.

Also, I could not distinguish how sequenced were groups in the methods, lines 161-178, so you should probably clarify that step. For example, some researchers cluster at 99% sequence identity rather than 100% and still refer to them as ASV.

Line 208. Be consistent when referring to normal at Dsc-affected sites (NAS). Throughout the text and within figures or captions, this sample type is referred to as NAS or Normal-AS or Normal at Affected Site. I recommend using Normal-AS throughout.

Lines 210-211. If any of the information included in Hewson et al., 2023 is relevant to this study, it should be written here as well.

Lines 229-230. I recommend breaking up this sentence following ...and coelomic fluid. Additionally, *Psychrobium* spp. Is a putative biomarker for DSc..."

Lines 233-236. Move the (Kruskal Wallis and Wilcoxon Test...) to the end of sentence following (Figure 3;...). It is easier to read

the sentence this way.

Line 238. Add "the" before genera.

Line 242. Can you add a transition statement to justify why you incorporated the challenge experiment into your study?

Line 257. Remove "doesn't". This is colloquial.

Lines 275-279. Can you report how many sequence libraries passed QC from the 13 urchin samples? See my comment for Figure 6.

Lines 300-302. That's cool!

Figure 1. Since there are many similar color hues, I recommend denoting the common genera with bold text or an asterisk. I had to search the text to figure out that the highly abundant seafoam green blocks pertained to *Kistimonas*, rather than *Candidatus_Riegeria*, *Cellulomonas*, *Halomonas*, *Pedobacter*, or R76-B128. Are the dark pink blocks *Psychrobium* or *Vibrio*? I also recommend that you append the Class designation to the numeric clades (e.g., 67-14, R7C24) and correct awkward genus designations like *P.palmC41*. It is frustrating that the alignment databases are sloppy with names, but this is your publication and I think you should make the graphics look as nice as possible.

Caption for Figure 1, Lines 595-597. Revise "except forwith the exception".

It is awkward to differentiate taxonomic assessment from phylogenetic assessment in the caption. I expect that Qiita assigns taxa based on multiple sequence alignment which is considered phylogenetic. I completely understand why you worked harder to better classify the *F. hongkongensis* using a tree-based method, but I recommend that you describe this process in the methods section, then leave it alone from there.

Figure 2 is an excellent graphic panel, nice job.

Figure 3 is problematic. Potentially the intended graphic got messed up during the .pdf conversion, but in my version the taxa names are distorted and unreadable. Also, it is unclear what the legend denoting Reference and Abnormal pertains to. Figure 4 does not offer enough information to warrant a graphic. I understand that you wish to illustrate that core bacteria were not shared between NAS, Abnormal, and Reference urchins. One option is just to describe this result using words. Another option is to combine the LDA and Venn diagram into Fig 3A, B. This would make sense because both graphics relate to the section: Comparison of grossly normal and DSc-affected urchin microbiomes in the Caribbean.

Figures 5 and 6. Same comment as with Figure 1 - please fix the names in your legend to include some type of taxonomic indicator (Class is always pleasant). Can you also fix your database so that *Fangia hongkongensis* is listed alphabetically like the other taxa?

Caption for Figure 6, Line 632. Fix ASVsASVs. Also, can you report how many good sequence libraries resulted from the Reunion Island urchins? Did all 13 libraries pass QC or does this reflect a reduced dataset?

We would like to thank all reviewers for their insightful and constructive reviews of the submitted version of this manuscript. We have made all the suggested changes to the manuscript to address these concerns, which are outlined below in red.

Reviewer #1 (Comments for the Author):

Overview

The study investigates the impact of *Diadema antillarum* scuticociliatosis (DSc), caused by the Philaster clade scuticociliate, on the microbiomes of Diadematidae sea urchins following mass mortality events in the Caribbean (Spring 2022) and subsequent outbreaks in the Eastern Mediterranean, Red Sea, and Western Indian Ocean. Using 16S rRNA gene amplicon sequencing, microbiome composition was compared between grossly normal and DSc-affected urchins from field samples and experimental challenges. Results show that unaffected urchins from reference sites have different representation and taxonomic richness of species than DSc-affected urchins. The findings suggest that DSc alters Diadematidae microbiomes. Some clarification is requested below.

Specific Comments

Abstract: It is perhaps too early to state that *Fangia hongkongensis* could be useful for early detection of *Diadema antillarum* scuticociliatosis as it is unknown at what stage of infection these bacteria colonize the urchin host.

We thank the reviewer for the comment, and we agree, we have adjusted the language of the sentence to reflect that this is a hypothesis and not a confirmation. For that matter the sentence now reads “Our results suggest that DSc alters Diadematidae microbiomes, and that *F. hongkongensis* may be a candidate bacterial biomarker...” (Line 41-43)

Lines 65-73: Additional interesting context for this disease is the recent efforts to understand factors limiting population recovery of this important grazer; one potentially relevant factor being the ongoing presence of a disease agent in populations.

We thank the reviewer for the suggestion and agree that some information should be detailed regarding current hypothesis of disease trajectory and recovery. We have added the following: “Moreover, *D. antillarum* recovery post mass mortality events have shown to be slow taking over a decade until natural recovery was seen on most reefs after the first mortality event in 1983. Currently it is hypothesized that due to the larger unaffected population of urchins in areas of the Caribbean the recovery should be substantially faster than before, but this remains to be seen (2).” (Line 78-81)

Lines 101-103: Also, the reverse: might bacterial communities affect risk of infection with ciliates?

Yes, we agree with the reviewer that it is a possibility that bacterial community could indeed influence the infection with ciliates, for this matter we added “Conversely, the host microbiome may modulate infection risk by ciliates, since ciliate infection can be correlated to disruptions of

the normal microbiome (26) and interruption of invertebrate microbiomes has been hypothesized to result in decreased host health (27).” (Line 104-107)

Lines 126-137: Why was sterile technique not applied across all types of samples collected?

All samples were collected using sterile technique, but we realize how the previous language made it seem as though that wasn't the case. We changed the language in the following sentence to make this clear. “Upon retrieval to the laboratory, urchins were necropsied into several tissue samples using sterile technique.” (Line 140-141)

Lines 290-299: Which hypothesis or combination of hypotheses do you find most likely?

Our data aids us in confirming these hypotheses and will form the basis for future work to address them. Despite this the data doesn't drive us to believe it is in favor of one hypothesis over another, which is why we believe more work needs to be done to support them.

Reviewer #2 (Comments for the Author):

Overall comments:

The study by Vilanova-Cuevas et al. describes the microbiomes associated with normal and abnormal (scuticociliatosis affected) urchins. The dataset represents a comprehensive sample collection of grossly normal and scuticociliatosis infected urchins (~300) from two major geographies, the Caribbean and the Indian Ocean. The study identified some interesting taxa that may serve as indicator species and point towards a functional understanding of the pathology. In my opinion, the study will make a good contribution to the field and is appropriate for the scope of mSystems. While the microbiome analysis was rigorous, there are several details that need to be included in the text and some of the graphics should be reconfigured. Detailed below are comments to improve both quality and clarity of the manuscript.

Specific comments:

Abstract:

Line 31, DScPc is never defined. Probably easiest to add the abbreviation to line 26. Also, the abstract uses abbrev. DaScPc while other sections use DScPc.

We agree with the reviewer and have made the changes so the abbreviation is included in Line 26 and have changed all abbreviations to be consistent across the manuscript.

Introduction:

Lines 59-60, For consistency, I recommend revising to: "Diadematidae scuticociliatosis (DSc) is a condition caused by a scuticociliate within the Philaster clade (abbreviated as DScPc)."

We agree and thank the reviewer for the suggestion and have made the revision as suggested in Line 61.

Methods:

Lines 179-190. Use this section to describe the name changes from *C. taeniosporalis* to *F.*

hongkongensis, etc. Then refrain from mentioning *Caedibacter* in the results text or figure captions because this is confusing.

While we understand the reviewer, we do believe the name change comes as a result from our analysis which makes it part of our result section and doesn't necessarily fit within the methods portion of the text. The figure legend has been changed to "Figure 1. Taxonomic assessment of bacterial genera from field collected samples that are above 5% relative abundance. The x-axis represents whether samples were grossly normal at reference site (Reference), grossly normal at affected site (Normal-AS) or DSc-affected (Abnormal). This represents the average abundance across samples pertaining to the same sample type and disease state, individual sample abundances are also detailed in Supplementary Figure 1."

Results:

Lines 196, 207, 231, etc. Capitalize Reference when referring to that specific sample type and fix throughout.

We agree with the reviewer and have made all necessary changes through the manuscript.

Lines 197-202. When you refer to ASVs, I believe this sequence grouping would represent one individual in the population, a sp. not spp. (which denotes several different species/strains within the genus). Unless you clustered sequences at 97% identity (OTUs), then it would be more appropriate to refer to the taxonomic group as spp.

Also, I could not distinguish how sequenced were groups in the methods, lines 161-178, so you should probably clarify that step. For example, some researchers cluster at 99% sequence identity rather than 100% and still refer to them as ASV.

We agree with the reviewer some clarity was needed in this area to explain the method efficiently. The taxonomy analysis was achieved by taking all the ASV's belonging to a same genus and clustering them together for this analysis purposes, so they are all spp.. We have now also added that the confidence threshold for assignment was 97% by adding "Taxonomy was assigned with 97% identity confidence threshold, using the Silva-138-99-515-806-nb-classifier (V.138) (Quast et al. 2013)" (Line 179-180)

Line 208. Be consistent when referring to normal at Dsc-affected sites (NAS). Throughout the text and within figures or captions, this sample type is referred to as NAS or Normal-AS or Normal at Affected Site. I recommend using Normal-AS throughout.

We agree that the category was inconsistently labeled across the manuscript, and we have fixed that now as suggested.

Lines 210-211. If any of the information included in Hewson et al., 2023 is relevant to this study, it should be written here as well.

We agree with the reviewer that the way it was written made it seem like more information was only available in the other manuscript, so we have edited to the following:

“Despite appearing healthy, NAS is grouped separately from reference urchins because these specimens may have had preclinical (i.e. before developing gross DSc signs) infection which may also have harbored altered microbiomes (1).” (Line 225-227)

Lines 229-230. I recommend breaking up this sentence following ...and coelomic fluid. Additionally, Psychrobium spp. Is a putative biomarker for DSc...”

We agree with the reviewer and have made the necessary changes as suggested. (Line 257-258)

Lines 233-236. Move the (Kruskal Wallis and Wilcoxon Test...) to the end of sentence following (Figure 3;...). It is easier to read the sentence this way.

We agree with the reviewer and have made the necessary changes as suggested. (Line 264)

Line 238. Add "the" before genera.

We agree with the reviewer and changes have been made as suggested. (Line 266)

Line 242. Can you add a transition statement to justify why you incorporated the challenge experiment into your study?

We have amended a transition statement that reads as follows “Comparing microbiome changes within in an experimental setting allows us to better understand if changes in microbial community directly correspond to that which happens in the environment and whether it influences disease outcome.” (Line 270-272)

Line 257. Remove "doesn't". This is colloquial.

We agree with the reviewer and have changed it to “... does not...” (Line 296)

Lines 275-279. Can you report how many sequence libraries passed QC from the 13 urchin samples? See my comment for Figure 6.

We agree this could have been clearer that all libraries worked well. We made the changes necessary as detailed here “Amplicon libraries, prepared from coelomic fluid and spines (n=13), had high representation of...” (Line 324)

Lines 300-302. That's cool!

Thank you!

Figure 1. Since there are many similar color hues, I recommend denoting the common genera with bold text or an asterisk. I had to search the text to figure out that the highly abundant seafoam green blocks pertained to Kistimonas, rather than Candidatus_Riegeria, Cellulomonas, Halomonas, Pedobacter, or R76-B128. Are the dark pink blocks Psychrobium or Vibrio? I also recommend that you append the Class designation to the numeric clades (e.g., 67-14,

R7C24) and correct awkward genus designations like P.palmC41. It is frustrating that the alignment databases are sloppy with names, but this is your publication and I think you should make the graphics look as nice as possible.

We agree with the reviewer and have made the necessary changes to this graph so that all the class names for those genera that are less intuitively named can be recognized. The names in the legend are listed in order as they appear on the bar which makes it as easy as possible to read given the large number of genera that are in the graph we believe adding asterisk would only make it more confusing.

Caption for Figure 1, Lines 595-597. Revise "except forwith the exception".

It is awkward to differentiate taxonomic assessment from phylogenetic assessment in the caption. I expect that Qiita assigns taxa based on multiple sequence alignment which is considered phylogenetic. I completely understand why you worked harder to better classify the *F. hongkongensis* using a tree-based method, but I recommend that you describe this process in the methods section, then leave it alone from there.

While we agree that it is unusual to describe this in the figure legend, we believe it appropriate. The reassignment occurred as part of the results of our phylogenetic analysis for that matter we believe adding it to the methods takes away from the results. Nonetheless we have edited the caption to fix the initial error pointed out so now it reads "Taxonomic assessment of bacterial genera, except for *Fangia hongkongensis* which had been initially..." (Figure 1 Caption)

Figure 2 is an excellent graphic panel, nice job.

Thank you!

Figure 3 is problematic. Potentially the intended graphic got messed up during the .pdf conversion, but in my version the taxa names are distorted and unreadable. Also, it is unclear what the legend denoting Reference and Abnormal pertains to.

Figure 4 does not offer enough information to warrant a graphic. I understand that you wish to illustrate that core bacteria were not shared between NAS, Abnormal, and Reference urchins. One option is just to describe this result using words. Another option is to combine the LDA and Venn diagram into Fig 3A, B. This would make sense because both graphics relate to the section: Comparison of grossly normal and DSc-affected urchin microbiomes in the Caribbean.

We fully agree with the reviewer and thank them for the suggestion to put both graphs together into one panel. We have moved forward with this suggestion and have made the necessary changes in the text to correspond to the figure.

Figures 5 and 6. Same comment as with Figure 1 - please fix the names in your legend to include some type of taxonomic indicator (Class is always pleasant). Can you also fix your database so that *Fangia hongkongensis* is listed alphabetically like the other taxa?

We have now added Class to all genera that are not intuitively labelled so that they can be easily identifiable. We don't think changing the database, so it is alphabetically labelled is particularly

relevant as it doesn't take away from the graph or results, the bacteria appear in the legend in the same top-to-bottom order as they do in the bar plot which facilitates readers identifying each bacterial group in order.

Caption for Figure 6, Line 632. Fix ASVsASVs. Also, can you report how many good sequence libraries resulted from the Reunion Island urchins? Did all 13 libraries pass QC or does this reflect a reduced dataset?

All changes have been addressed as suggested and clarifications of the number of libraries were addressed in Line 324.

Re: mSystems01418-24R1 (Investigating the influence of Diadematidae scuticociliatosis on host microbiome composition)

Dear Dr. Brayan Vilanova-Cuevas:

Your manuscript has been accepted, and I am forwarding it to the ASM production staff for publication. Your paper will first be checked to make sure all elements meet the technical requirements. ASM staff will contact you if anything needs to be revised before copyediting and production can begin. Otherwise, you will be notified when your proofs are ready to be viewed.

Sincerely,
Jean-Baptiste Raina
Editor
mSystems

Reviewer #1 (Comments for the Author):

The authors have appropriately addressed my previous comments.

Reviewer #2 (Comments for the Author):

Thank you for your careful consideration of the reviewer comments.